# Rethinking DeepVariant: Efficient Neural Architectures for Intelligent Variant Calling

**DOI:** 10.3390/ijms27010513

**Published:** 2026-01-04

**Authors:** Anastasiia Gurianova, Anastasiia Pestruilova, Aleksandra Beliaeva, Artem Kasianov, Liudmila Mikhailova, Egor Guguchkin, Evgeny Karpulevich

**Affiliations:** 1Ivannikov Institute for System Programming of the Russian Academy of Science, 109004 Moscow, Russia; a.gurianova@ispras.ru (A.G.);; 2Center for Applied AI, Skolkovo Institute of Science and Technology, 121205 Moscow, Russia; 3BIOPOLIS Program in Genomics, Biodiversity and Land Planning, CIBIO, 4485-684 Vairão, Portugal; 4Economic Faculty, Lomonosov Moscow State University, 119991 Moscow, Russia; 5Higher School of Management, Financial University Under the Government of the Russian Federation, 125167 Moscow, Russia; 6Research Center for Trusted Artificial Intelligence, Ivannikov Institute for System Programming of the Russian Academy of Science, 109004 Moscow, Russia

**Keywords:** DeepVariant, variant calling, SNP, convolutional neural networks, genomics, GIAB, WGS, WES, NGS

## Abstract

DeepVariant has revolutionized the field of genetic variant identification by reframing variant detection as an image classification problem. However, despite its wide adoption in bioinformatics workflows, the tool continues to evolve mainly through the expansion of training datasets, while its core neural network architecture—Inception V3—has remained unchanged. In this study, we revisited the DeepVariant design and presented a prototype of a modernized version that supports alternative neural network backbones. As a proof of concept, we replaced the legacy Inception V3 model with a mid-sized EfficientNet model and evaluated its performance using the benchmark dataset from the Genome in a Bottle (GIAB) project. Alternative architecture demonstrated faster convergence, a twofold reduction in the number of parameters, and improved accuracy in variant identification. On the test dataset, updated workflow achieved consistent improvements of +0.1% in SNP F1-score, enabling the detection of up to several hundred additional true variants per genome. These results show that optimizing the neural architecture alone can enhance the accuracy, robustness, and efficiency of variant calling, thereby improving the overall quality of sequencing data analysis.

## 1. Introduction

Whole-genome sequencing (WGS) and whole-exome sequencing (WES) generate millions to billions of reads per sample, enabling more sensitive and cost-effective disease diagnostics than those of traditional screening methods [1,2,3,4,5,6]. Accurate detection of genetic variants—single-nucleotide variants (SNVs) and indels—is a critical step, known as variant calling, required to translate these reads into clinically relevant information. Early and reliable variant identification supports risk assessment and informs treatment decisions [7]. Conversely, calling errors incur immediate clinical and economic costs: false positives necessitate confirmatory tests, while false negatives can lead to diagnostic failure or inappropriate therapy [8]. Current clinical guidelines identify high-accuracy variant calling as a critical bottleneck in the genome analysis pipeline [9].

Traditional variant calling methods rely on statistical models and heuristic algorithms. In contrast, DeepVariant approaches variant calling nontrivially as an image-classification problem solved by deep convolutional neural networks, widely adopted in medicine and biology over the past decade [10,11,12,13]. This makes DeepVariant one of the widely used applications employing image-based data representations, together with deep learning models, for genome analysis, alongside approaches such as CGR image-encoded analysis and integrative spatial gene-expression imaging models [14]. Multiple studies confirm DeepVariant superiority over well-established tools such as Genome Analysis Toolkit (GATK) and Strelka in terms of accuracy and recall [15,16]. Its advantage over statistics-based approaches was demonstrated in the PrecisionFDA Truth Challenge, solidifying its status as one of the most accurate variant callers [15].

The DeepVariant pipeline comprises three stages: (i) ‘make_examples’—aligned reads around putative variants (BAM) are encoded as multi-channel images; (ii) ‘call_variants’—CNN classifies each image into one of three classes reflecting diploid genotype states: 0/0 (homozygous reference: both alleles match reference), 0/1 (heterozygous: one reference allele and one alternate), or 1/1 (homozygous alternate: both alleles differ from reference); and (iii) ‘postprocess_variants’—CNN outputs are converted to standardized VCF records [17]. These stages are summarized in Figure 1.

Continuous accumulation of genomic data and the emergence of new sequencing technologies have driven significant evolution in DeepVariant since its initial release. Most improvements have focused on expanding the volume and diversity of training data and adapting to new sequencing techniques, including extension from WGS/WES to RNA-Seq [16] and support for third-generation platforms such as PacBio and Oxford Nanopore [18,19].

Modifications to the pipeline itself have been largely confined to the ‘make_examples’ stage, where variant context was enriched by adding a haplotype channel during pileup image encoding. The concept of read haplotagging was first explored by Shafin et al., who proposed the PEPPER-Margin-DeepVariant pipeline for nanopore long reads, and by Patterson et al., who showed that haplotype information improves variant detection accuracy with PacBio long reads [18]. Early implementations used Margin and WhatsHap for haplotagging [20,21]. This approach was later refined by Kolesnikov et al., who introduced an approximate haplotagging method that simplifies long-read variant calling [17]. Haplotype information is now included as a default channel in the DeepVariant pipeline.

Despite these successful modifications, DeepVariant backbone CNN—Inception-v3, used in the ‘call_variants’ stage—has remained unchanged since the early releases [22]. In the intervening years, the field of computer vision has advanced substantially. Progress has included improved training of very deep networks through residual connections in ResNet [23] and more efficient feature utilization and compound scaling strategies in models like EfficientNet [24]. Beyond these advancements, computer vision models have increasingly embraced hybrid and fully transformer-based designs. Integration of architectural insights from transformers exemplified by modernized CNNs like ConvNeXt [25,26] and Vision Transformers (ViT) demonstrated that, with sufficient data, self-attention can rival or surpass convolutional architectures for image classification [27]. Subsequent refinements—such as DeiT [28], which introduced more data-efficient training strategies, and Swin Transformer, which added hierarchical feature maps and shifted windows—facilitated the transition of transformers into fields requiring fine-grained, spatially aware predictions [29].

The continued use of Inception V3 within DeepVariant is a de facto standard. While most researchers enhance the pipeline through fine-tuning or adding new input channels, the fundamental question, ‘What if we replace the neural network backbone itself?’ has received comparatively little attention. Several community efforts have experimented with alternative architectures, but have typically focused on domain-specific modifications and yet none provide a direct, head-to-head comparison against DeepVariant’s original Inception V3 model within the full training pipeline [30,31]. In this work, we extend the official DeepVariant source code to enable the seamless substitution of alternative neural network backbones, allowing systematic exploration beyond the default Inception V3 architecture. To demonstrate this capability, we selected a representative alternative model and trained it from scratch alongside the baseline model under identical conditions. Performance was evaluated using a GIAB benchmark dataset [32], enabling a direct and reproducible comparison between backbone architectures.

## 2. Results

### 2.1. Training

Baseline Inception V3 and a preselected alternative model were trained and evaluated over 10 independent cross-validation runs. F1 was the primary evaluation metric, defined as the harmonic mean of precision and recall, which were also reported as supplementary metrics. Per-class (stratified) performance analyses were provided.

#### 2.1.1. Learning Dynamics

Both baseline and alternative models exhibit a rapid loss decrease during early training steps, indicating quick learning, followed by stabilization and convergence (Figure 2). EfficientNet-B3 achieves lower training loss, demonstrating superior fit to the training data. Validation loss tends to plateau more gradually, while mirroring the training loss trend. On validation, EfficientNet-B3 once again attains lower loss, suggesting better generalization to the unseen validation set with a substantial performance gap relative to the baseline model.

#### 2.1.2. Performance Evaluation

We further analyze trajectories of the F1 score, precision and recall across training epochs for the aforementioned models (Figure 3). The F1 curves demonstrate a consistent separation in favor of the alternative EfficientNet model, indicating overall performance improvement. Given the already high precision of the baseline model, the gain achieved by the alternative model is comparatively modest. In contrast, the recall curves reveal a persistent and credible advantage—an approximately one percentage-point increase—suggesting that the observed improvement in the F1 score is primarily driven by improved sensitivity.

The alternative architecture outperforms baseline Inception_V3 across all key metrics at the final epoch evaluation, as shown in Table 1, indicating an improved capacity to classify variants. The EfficientNet-B3 model achieves a higher overall F1 score of 95.31%, representing a 0.51 percentage-point improvement over the baseline 94.80%, with a significant improvement (paired *t*-test *p* = 0.0067) and a large effect size (Cohen’s d = 1.71). The alternative model achieves a precision of 95.78%, surpassing the strong baseline of 95.63% by 0.15 percentage points; however, this difference is not statistically significant (*p* = 0.243, Cohen’s d = 0.40). Most notably, the recall improves to 94.78%, marking a substantial 1.02 percentage-point gain over the baseline 93.76%, with a notably large effect size (*p* = 0.0003, Cohen’s d = 1.79). Collectively, these results provide evidence that EfficientNet-B3 is more effective at correct identification of true variants.

#### 2.1.3. Performance Stratified by Genotype

We next conducted a stratified evaluation of validation metrics across three variant classes: homozygous reference (0/0), heterozygous (0/1), and homozygous alternate (1/1), as presented on Figure 4. This analysis enables the assessment of whether the models exhibit any class-specific preferences.

For F1 scores, the alternative model demonstrates consistent improvement over the baseline across all variant classes. Furthermore, the overall performance pattern reflects that of the baseline: the classification is most accurate for genotypes with two alternate alleles (homozygous alternate, 1/1), followed by those with a single alternate allele (heterozygous, 0/1), while the greatest challenge is observed in correctly identifying the homozygous reference (0/0), despite the fact that these variants were the most frequent in the training dataset.

However, when examining precision and recall separately, we noticed more nuances. The baseline model exhibits consistently low sensitivity across all genotype classes. The alternative model follows a similar trend for the most challenging homozygous reference (0/0) class. In contrast, for the other two classes (heterozygous, 0/1, and homozygous alternate, 1/1), the pattern reverses, with recall increasing relative to precision. This highlights the potential class-specific strengths and limitations of the EfficientNet architecture in genotype prediction, which remained obscured in aggregate performance metrics.

### 2.2. Testing

Trained models were included in the original DeepVariant pipeline to conduct a final evaluation of two independent test sets from hold-out samples HG003 and HG005 using hap.py (v0.3.12). Results are summarized separately for SNP and indel variants.

For SNP, both models demonstrate very high performance, with F1 scores exceeding 99% (Table 2 and Table 3). On the never-seen samples—belonging to a distinct ancestry with a divergent genotype class distribution—performance improvement of the alternative model persists. Across both HG003 and HG005, the DeepVariant pipeline incorporating EfficientNet-B3 consistently improves performance. For HG003, the model yields a 0.10 percentage-point increase in F1 (*p* = 0.0107, Cohen’s d = 1.68), accompanied by a 0.08 point gain in precision (*p* = 0.0471, Cohen’s d = 0.94) and a 0.11 point increase in recall (*p* = 0.1708, Cohen’s d = 0.69). Similarly, for HG005, F1 improves by 0.07 percentage points (*p* = 0.0075, Cohen’s d = 1.75), precision by 0.08 points (*p* = 0.0372, Cohen’s d = 1.00), and recall by 0.07 points (*p* = 0.2423, Cohen’s d = 0.57). Together, these results demonstrate consistent gains across metrics, with statistically significant improvements in F1 and precision for both samples and modest non-significant enhancements in recall.

For indel classification—which has traditionally posed a more challenging scenario—both models demonstrate modest performance, with F1 scores approaching 95%. For HG003, EfficientNet-B3 yields a modest 0.23 percentage-point improvement in F1 (*p* = 0.6301, Cohen’s d = 0.27). Precision shows a clearer benefit, increasing by 0.18 points (*p* = 0.0042, Cohen’s d = 1.61), while recall improves by 0.29 points, although without statistical significance (*p* = 0.7455, Cohen’s d = 0.18). A similar pattern is observed for HG005. F1 improves by 0.19 percentage points (*p* = 0.3877, Cohen’s d = 0.46), precision increases by 0.23 points, with strong statistical support (*p* = 0.0022, Cohen’s d = 1.86), and recall shows a small, non-significant gain of 0.15 points (*p* = 0.7070, Cohen’s d = 0.20).

In the context of indel detection, which is considerably less represented in the training data, the proposed model shows clear improvements in precision; however, gains in recall (and consequently, sensitivity) remain challenging. In HG003, for example, recall is nearly 10 percentage points lower than precision, underscoring the difficulty the model faces in recovering true indel variants. Although mean recall and F1 values trend upward across independent runs, the high variability observed for indels prevents drawing confident conclusions about consistent improvements in recall and downstream increases in F1 overall.

### 2.3. Training Efficiency and Inference Time

Training times varied across models, reflecting differences in architectural complexity and computational efficiency (Table 4). For a 10-epoch training run on chromosome 10 of the HG001 sample (approximately 350,000 variants, 30× WGS), the baseline model required, on average, 2 h and 34 min, whereas the alternative model completed training in 1 h and 59 min. This demonstrates that the alternative model achieves faster convergence, offering advantages in both accuracy and training efficiency. During inference, however, both models performed similarly: the full DeepVariant pipeline for the HG003 and HG005 samples (excluding chromosomes used in training to prevent leakage) required 2 h per one cross validation run, indicating that inference speed is largely unaffected by the model architecture.

## 3. Discussion

### 3.1. Alternative Model Performance

The alternative mid-sized EfficientNet model demonstrates substantial and consistent advantages over baseline Inception V3 across multiple evaluation dimensions.

During training, losses were markedly lower for EfficientNet-B3, indicating improved fitting to the data and better generalization. The alternative model exhibited more stable learning dynamics and reduced overfitting, reflecting architectural efficiency in extracting relevant features for variant classification. The performance gap becomes apparent early during training and remains stable after convergence, reflecting the architectural robustness and effective feature learning of EfficientNet-B3.

Throughout training, EfficientNet-B3 consistently outperforms the Inception V3 baseline across all key metrics. Curve analysis shows a clear separation in F1 curves; by the final epoch, EfficientNet-B3 achieves a mean F1 score of 95.31% (+0.51%), highlighting its enhanced classification performance. While both models exhibit high precision, EfficientNet-B3 shows improved recall of 94.78% (+1.02%), indicating that the overall performance gain is primarily driven by better sensitivity. These findings highlight EfficientNet-B3’s superior ability to detect true variants.

Stratified evaluation confirms that EfficientNet-B3 consistently outperforms the Inception V3 baseline across all genotype classes. While both models follow a similar F1 score pattern—best for the homozygous alternate (1/1), then the heterozygous (0/1), and worst for the homozygous reference (0/0)—EfficientNet-B3 achieves higher scores throughout. An analysis of precision and recall reveals further differences. Inception V3 adopts a conservative prediction strategy, resulting in uniformly low sensitivity. EfficientNet-B3 partially retains this pattern for the homozygous reference (0/0) class but shifts toward higher recall for genotypes with alternate alleles. This suggests improved detection of true variants, even despite their lower prevalence in the training set.

In the final evaluation, trained models were integrated into the original DeepVariant pipeline and tested on two held-out WGS samples, HG003 and HG005, using hap.py. Across both datasets, the EfficientNet-B3 architecture consistently outperformed the baseline Inception V3 model. For SNPs, performance was uniformly high, with F1 scores exceeding 99%, and the alternative model providing statistically significant gains in regards to both F1 and precision. Indel classification, while more challenging, showed a similar pattern. EfficientNet-B3 delivered clear and statistically supported improvements in precision for both samples, whereas gains in recall were small and variable. This reflects the difficulty the proposed model faces in enhancing indel sensitivity—particularly evident in HG003, where recall remains nearly 10 percentage points lower than precision for both models. Although average recall and F1 values trend positively across independent runs, high variability limits confidence in consistent improvements for these metrics. Overall, results indicate that modernized architecture provides robust and reproducible gains, especially through precision improvements, while highlighting specific areas—most notably indel recognition—where further optimization remains needed.

Notably, these performance gains are achieved alongside substantial improvements in model efficiency. EfficientNet-B3 displays approximately half as many parameters as the baseline Inception V3, leading to reduced GPU usage and faster training time. These efficiency advantages reflect the architectural strengths of EfficientNet, which is designed around a compound scaling strategy that balances network depth, width, and resolution in a principled manner. This allows the model to capture fine-grained patterns in pileup images more effectively, which is crucial for distinguishing subtle variant signals.

Importantly, in this study, we employed a relatively lightweight configuration; more complex architectures could potentially allow even greater gains, particularly in more challenging scenarios on larger-scale datasets. Together, these findings suggest an alternative trajectory for improving DeepVariant performance—focusing on architectural modernization rather than relying solely on scaling with ever-larger volumes of training data.

In this context, we present a case that we hope will contribute to rethinking current practices in genomic variant classification pipelines: leveraging well-designed models may offer a more sustainable and scalable path forward than will continued reliance on computationally intensive legacy architectures.

### 3.2. Limitations

While these results are promising, several limitations must be acknowledged. The current study is restricted to a subset of Illumina whole-genome sequencing (WGS) samples (2 × 151 bp, 30× coverage), which were preselected as representative cases to enable fast prototyping. Also, these findings may not fully generalize to other sequencing setups, platforms based on long-read technologies (e.g., PacBio or Oxford Nanopore), or low-coverage libraries (≤15×), which are known to exhibit distinct error profiles and could alter the relative performance of the evaluated models [33,34].

### 3.3. Future Directions

Future work may explore several directions. Incorporating advanced read alignment methods—already shown to enhance SNV detection accuracy—could further improve classification outcomes [35]. Additionally, experimenting with alternative pileup image encoding strategies, integrating supplementary input channels, or applying data augmentation techniques may enrich the feature space and model generalization. Finally, expanding the training dataset remains a logical next step, although the computational and financial cost of large-scale model training presents a significant practical challenge.

## 4. Materials and Methods

### 4.1. Datasets

DeepVariant production models were trained and evaluated using a well-defined GIAB dataset. This dataset includes high-confidence variant calls for multiple individuals and serves as a standard benchmark for variant calling. Specifically, seven samples from different individuals were used in the original DeepVariant setup. Six samples (HG001, HG002, HG004, HG005, HG006, HG007) were used for training (chromosomes 1–19 for training itself; chromosomes 21–22 for validation and evaluation of performance during training), while one sample (HG003, chromosome 20) was kept for testing [12].

In our study, we followed the same methodological principles but in a reduced configuration. Training is the most resource-demanding stage, and training full production-scale models for each architectural candidate is computationally prohibitive; therefore, we adopted a simplified GIAB setup consistent with the DeepVariant advanced training tutorial (https://github.com/google/deepvariant/blob/r1.6.1/docs/deepvariant-training-case-study.md, accessed on 10 September 2025) to rapidly prototype and evaluate the proposed models. In designing the dataset, we aimed to ensure diverse samples, as well as complete separation between the training, validation, and testing regions. We used BAM files from the original article’s supplementary materials, kindly provided by the DeepVariant authors [36], corresponding to NovaSeq 30× whole-genome sequencing, a standard and well-characterized coverage for WGS applications (sample details in Table 5). For model development, HG001 chromosome 10 and HG001 chromosome 20 were used for training and validation, respectively. Chromosomes 10 and 20 were selected because they provide representative genome regions of moderate size and variant density, making them suitable for rapid iteration during model development while still reflecting the characteristics of whole-genome variant calling tasks. For the final evaluation, we used the whole HG003 and HG005 samples, excluding the aforementioned regions to prevent data leakage.

Additionally, we present the distribution of variant classes across datasets. As shown in Table 6, in the training and validation datasets (HG001), homozygous reference variants (0/0) predominate, followed by heterozygous (0/1) and homozygous alternate (1/1). This distribution differs from that of the test set, where heterozygous variants represent the largest proportion among the three classes.

### 4.2. Selection of Alternative Model for Demonstration

Selection of the alternative model was guided by reference information about models from Keras Applications (https://keras.io/api/applications/ accessed on 27 November 2025). We aimed to balance accuracy and computational efficiency. Among the wide range of alternative architectures—including the previously mentioned popular choices—we focused on the EfficientNet family. EfficientNet models employ a compound-scaling strategy that jointly increases network depth, width, and input resolution in a balanced and principled manner. This design achieves substantially better accuracy-parameter trade-offs than that of earlier architectures, including Inception V3. For pileup tensors—which are smaller and structurally more constrained than natural images—such efficiency is particularly advantageous, enabling the model to capture relevant variation patterns without incurring unnecessary depth or computational cost. Additionally, variant-calling signals frequently appear as subtle local features (base-quality gradients, for example), and EfficientNets’ MBConv blocks with squeeze-and-excitation mechanisms enhance the network’s sensitivity to these fine-grained cues, even when the available global context is limited.

Within the EfficientNet family, we selected the mid-sized EfficientNet-B3 model (Figure 5). Model capacity and computational cost grow rapidly across the EfficientNet scale indices (B0–B7) due to the compound-scaling strategy, and preliminary experiments on public benchmark ImageNet dataset indicated that larger variants (B4–B7) provide only modest accuracy improvements while incurring disproportionately higher memory usage and training time. In contrast, EfficientNet-B3 offered a favorable balance: it delivered substantial accuracy gains over those of Inception V3, while maintaining a model size and inference cost that are compatible with the practical DeepVariant training pipeline. This combination of improved accuracy and moderate resource demands made EfficientNet-B3 a compelling and practically viable candidate for integration into DeepVariant.

### 4.3. Training

#### 4.3.1. Training Pipeline Adaptation

Most changes were made in keras_modeling.py. Key modifications include the introduction of a configurable _BACKBONES dictionary within the model codebase, enabling selection among multiple neural network architectures—Inception V3 and EfficientNet B0-B7—accessed via the tf.keras.applications module, with the potential for future extension to other architectures. The get_model() function was adapted to instantiate the chosen backbone architecture, defaulting to Inception V3, and configured to accept input tensors representing pileup image data. Additionally, logging was updated throughout to improve training reproducibility. The train.py script logic remained unchanged, but we integrated support for other optimizers such as Adam and AdamW.

#### 4.3.2. Training Configurations

In the original pipeline configurations—including parameters such as batch size, learning rate, optimizer, logging frequency, and others—are specified through the get_config() function in dv_config.py. We updated this script to allow the selection of different model architectures (e.g., EfficientNet-B0/B1/B2).

Configurations for the baseline Inception V3 model were based on the original script. In our setup, the batch size was reduced from 16,384 to 128 to match the limitations of the hardware. Additionally, as we trained models from scratch with randomly initialized weights, pre-trained model checkpoints were disabled. The baseline model was trained for 10 epochs using the RMSProp optimizer with a momentum of 0.9, an initial learning rate of 1 × 10^−3^ with exponential decay (decay rate of 0.947) every 2 epochs, a weight decay of 4 × 10^−5^, and a dropout in the backbone network with a probability of 0.2 following the original script. Model evaluation (validation) was performed obligatorily at the end of every epoch. Model checkpoints were saved based on the best validation F1 score.

To train an alternative model, we used the same settings but made a preliminary search for the optimal learning rate and the optimizer via automated hyperparameter optimization with Optuna [37]. A Tree-Structured Parzen Estimator (TPE) sampler guided the search across 20 trials, and the Successive Halving algorithm was implemented to terminate underperforming trials early.

#### 4.3.3. Evaluation of Performance During Training

Model performance was evaluated primarily using the F1 score, as in the original article. Additionally, precision and recall were reported. To assess stability, 10 independent runs per model were implemented, and a 95% confidence interval was estimated using standard error and t-distribution. In addition, statistical significance between models was evaluated using *p*-values, and the magnitude of performance differences was quantified using Cohen’s d. The results were obtained using the services of the Shared Resource Center of the Ivannikov Institute for System Programming of the Russian Academy of Sciences – the ISP RAS Shared Resource Center.

### 4.4. Testing

Trained models were integrated into the full DeepVariant pipeline. The resulting variant calls (VCF files) were evaluated using hap.py, a standard tool for benchmarking variant callers [38]. Performance was assessed separately for SNPs and indels across all 10 independent runs. Statistical significance between models was evaluated using *p*-values, and the magnitude of performance differences was quantified using Cohen’s d.

### 4.5. Hardware and Environment

Model training and testing were performed on an NVIDIA A100 GPU with 40 GB of memory. The software environment was built using a custom container based on the official google/deepvariant:1.6.1-gpu Docker image.

### 4.6. Code Availability

The source code is publicly available in the repository https://github.com/ispras/deepvariant_alternative_models, accessed on 27 November 2025.

## 5. Conclusions

This study demonstrates that a mid-sized EfficientNet architecture provides a substantial and consistent improvement over the widely used Inception_V3 backbone for genomic variant classification within the DeepVariant pipeline. Across the training, validation, and evaluation stages, EfficientNet-B3 exhibits more stable learning dynamics, reduced overfitting, and markedly lower losses, indicating superior feature extraction and stronger generalization. These advantages manifest early during optimization and persist through convergence.

EfficientNet-B3 consistently outperforms the baseline across all principal metrics during training, with F1 curves showing a clear and sustained separation. Stratified analyses further confirm greater robustness across genotype classes. Independent evaluation on held-out WGS samples (HG003 and HG005) reinforces these findings. For SNP classification, EfficientNet-B3 provides statistically significant improvements in regards to both F1 and precision. Indel classification—traditionally more difficult—shows reliable and statistically supported precision improvements, although gains in recall and subsequently F1 are highly variable and limited. These trends highlight both the strengths and current limits of the architecture, pointing to indel sensitivity as a key target for future refinement.

Beyond accuracy, EfficientNet-B3 delivers notable efficiency benefits. With roughly half the parameters of Inception V3, the model reduces GPU memory requirements and shortens training time, reflecting the advantages of compound scaling for balancing network depth, width, and resolution. These architectural efficiencies enable the extraction of fine-grained patterns in pileup images more effectively without incurring additional computational burden.

Taken together, the results illustrate that thoughtful architectural modernization can yield meaningful advances in variant classification. The success of a relatively lightweight EfficientNet-B3 configuration suggests that more advanced architectures may unlock even greater improvements.

## Figures and Tables

**Figure 1 ijms-27-00513-f001:**
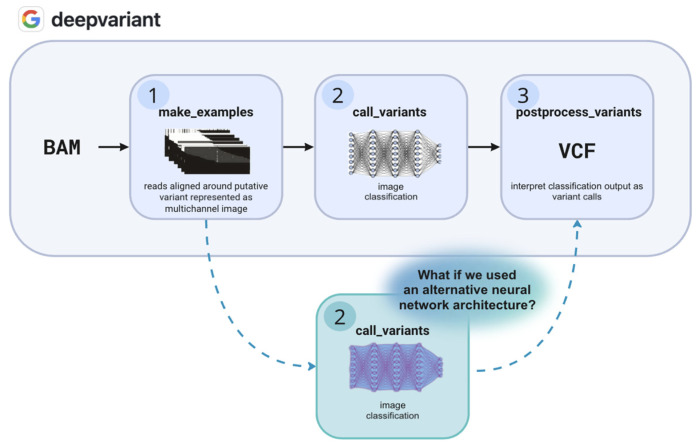
Overview of the DeepVariant variant-calling pipeline, which consists of three main steps: ‘make_examples’, ‘call_variants’, and ‘postprocess_variants’. Our proposed modification involves replacing the baseline CNN architecture in the classification step.

**Figure 2 ijms-27-00513-f002:**
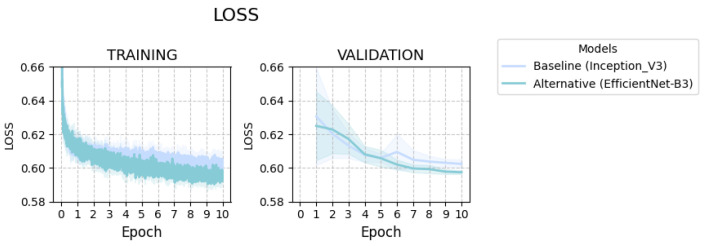
Training (**left** panel) and validation (**right** panel) loss curves for the baseline Inception_V3 model and the alternative EfficientNet-B3 model. Each line represents the mean across 10 independent runs, and nearby shaded area indicates the confidence interval (CI).

**Figure 3 ijms-27-00513-f003:**
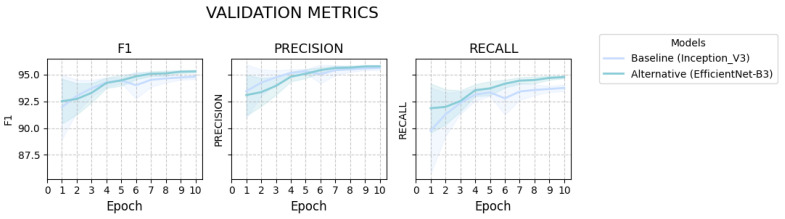
Progression of F1 score (**left** panel), precision (**middle** panel), and recall (**right** panel) throughout the validation steps for the baseline Inception_V3 and alternative EfficientNet-B3 model. Each line represents the mean across 10 independent runs, and the nearby shaded area indicates the confidence interval (CI).

**Figure 4 ijms-27-00513-f004:**
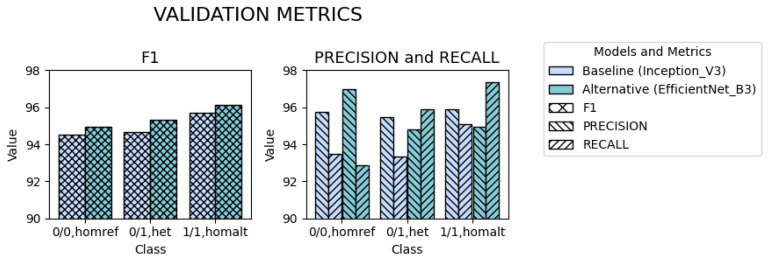
Stratified validation performance across genotype classes. Values are reported as mean across independent runs. Different colors represent baseline Inception V3 and alternative EfficientNet-B3 models; hatch patterns correspond to validation metrics: F1 score, precision, and recall.

**Figure 5 ijms-27-00513-f005:**
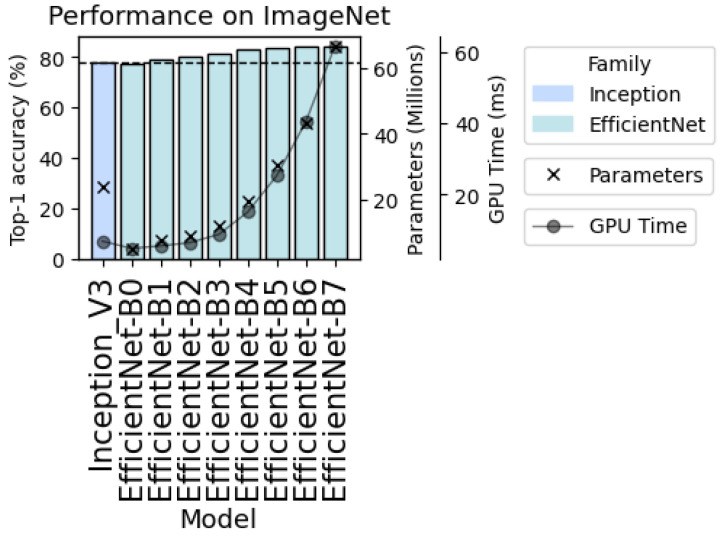
Comparison of baseline Inception V3 and EfficientNet models in terms of accuracy, parameter count, and GPU training time. Bars represent top-1 accuracy (%); dots connected by a line indicate GPU training time; crosses denote number of parameters. The selected EfficientNet-B3 model represents a preferable balance between accuracy gain, as well as a lower number of parameters and training/inference time.

**Table 1 ijms-27-00513-t001:** Performance comparison of baseline Inception_V3 and EfficientNet-B3 models on the validation set. Values are reported as the mean across independent runs with 95% confidence intervals (CI), paired *t*-test *p*-values, and Cohen’s d effect sizes. ↑ indicates superior results across evaluated metrics; * indicates statistically significant results (*p* < 0.05).

Metric	Baseline (Inception_V3)	Alternative (EfficientNet_B3)	*p*-Value	Cohen’s d
F1	94.8 (95% CI: 94.56–95.05)	↑ 95.31 (95% CI: 95.2–95.42)	0.0067 *	1.88
Precision	95.63 (95% CI: 95.41–95.84)	↑ 95.78 (95% CI: 95.69–95.88)	0.2432	0.67
Recall	93.76 (95% CI: 93.44–94.08)	↑ 94.78 (95% CI: 94.64–94.91)	0.0003 *	2.96

**Table 2 ijms-27-00513-t002:** Performance comparison of baseline Inception V3 and alternative EfficientNet-B3 on independent test sample HG003. Values are reported as the mean across independent runs with 95% confidence intervals (CI), paired *t*-test *p*-values, and Cohen’s d effect sizes. ↑ indicates superior results across evaluated metrics; * indicates statistically significant results (*p* < 0.05).

Metric	Baseline (Inception_V3)	Alternative (EfficientNet_B3)	*p*-Value	Cohen’s d
SNP				
F1	99.04 (95% CI: 99.01–99.08)	↑ 99.14 (95% CI: 99.1–99.18)	0.0107 *	1.68
Precision	99.64 (95% CI: 99.57–99.71)	↑ 99.72 (95% CI: 99.67–99.78)	0.0471 *	0.94
Recall	98.45 (95% CI: 98.36–98.55)	↑ 98.56 (95% CI: 98.44–98.68)	0.1708	0.69
Indel				
F1	93.8 (95% CI: 93.2–94.4)	↑ 94.03 (95% CI: 93.38–94.69)	0.6301	0.27
Precision	98.83 (95% CI: 98.74–98.92)	↑ 99.01 (95% CI: 98.94–99.07)	0.0042 *	1.61
Recall	89.26 (95% CI: 88.2–90.32)	↑ 89.55 (95% CI: 88.36–90.73)	0.7455	0.18

**Table 3 ijms-27-00513-t003:** Performance comparison of baseline Inception V3 and alternative EfficientNet-B3 on independent test sample HG005. Values are reported as the mean across independent runs with 95% confidence intervals (CI), paired *t*-test *p*-values, and Cohen’s d effect sizes. ↑ indicates superior results across evaluated metrics; * indicates statistically significant results (*p* < 0.05).

Metric	Baseline (Inception_V3)	Alternative (EfficientNet_B3)	*p*-Value	Cohen’s d
SNP				
F1	99.11 (95% CI: 99.07–99.14)	↑ 99.18 (95% CI: 99.15–99.22)	0.0075 *	1.75
Precision	99.64 (95% CI: 99.57–99.71)	↑ 99.72 (95% CI: 99.67–99.78)	0.0372 *	1.00
Recall	98.58 (95% CI: 98.5–98.66)	↑ 98.65 (95% CI: 98.55–98.75)	0.2423	0.57
Indel				
F1	97.1 (95% CI: 96.82–97.38)	↑ 97.29 (95% CI: 96.99–97.59)	0.3877	0.46
Precision	99.01 (95% CI: 98.9–99.12)	↑ 99.24 (95% CI: 99.17–99.3)	0.0022 *	1.86
Recall	95.27 (95% CI: 94.78–95.76)	↑ 95.42 (95% CI: 94.85–95.98)	0.7070	0.20

**Table 4 ijms-27-00513-t004:** Performance comparison of baseline Inception_V3 and alternative EfficientNet-B3 on training. Values are reported as top-1 accuracy (%) regarding ImageNet, number of parameters, training time.

Metric	Baseline (Inception_V3)	Alternative (EfficientNet_B3)
Acc@1	77.9	81.6
Params	23.9 M	12.3 M
Training time	2 h and 34 min × 10 CV runs	1 h and 59 min × 10 CV runs

**Table 5 ijms-27-00513-t005:** Characteristics of GIAB samples used in this study.

Sample	Gender	Ancestry	BAM File
HG001	Female	Utah/ European Ancestry	HG001.novaseq.pcr-free.30x.dedup.grch38.bam
HG003	Male	Eastern European Ashkenazi Jewish Ancestry	HG003.novaseq.pcr-free.30x.dedup.grch38.bam
HG005	Male	Chinese Ancestry	HG005.novaseq.pcr-free.30x.dedup.grch38.bam

**Table 6 ijms-27-00513-t006:** Distribution of genotype classes across datasets, samples, and chromosomes.

Dataset	Sample	Chromosomes	Number of Variants			
			Total	0/0, Homozygous Reference	0/1, Heterozygous	1/1, Homozygous Alternate
Training	HG001	10	355,674	153,737	123,611	78,326
Validation	HG001	20	156,159	71,858	53,263	31,038
Test 1	HG003	Other	4,822,356	1,196,591	2,196,336	1,429,429
Test 2	HG005	Other	4,473,118	1,008,100	1,975,012	1,490,006

## Data Availability

The datasets used in this study are publicly available and can be accessed through the original source cited in the manuscript.

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
