# Peer review of "Rethinking DeepVariant: Efficient Neural Architectures for Intelligent Variant Calling"

_ijms, 2026, doi:10.3390/ijms27010513_

Round 1
Reviewer 1 Report
Comments and Suggestions for Authors
Comment 1:
Regarding reproducibility, the original DeepVariant training pipeline uses seven samples from different individuals and incorporates multiple chromosomes to avoid overfitting. In contrast, the present study reduces the dataset to only two samples. The manuscript does not sufficiently justify why this downscaling was necessary, why these specific two samples were selected, and whether such selection introduces potential bias. It is also unclear whether this reduced dataset is representative of large-scale scenarios and whether the conclusions generalize across diverse populations and sequencing conditions.
Comment 2:
The Introduction lacks adequate comparison with recent advances in neural network architectures. For instance, modern CNNs such as ConvNeXt and other contemporary vision backbones are not discussed. A more thorough review of related work is needed to contextualize the contribution.
Comment 3:
The manuscript only compares Inception V3 with EfficientNet-B3, making it difficult to support claims of generalizability. It would be helpful to include a comparison table summarizing model size, number of parameters, training time, and performance across different architectures to provide a more comprehensive evaluation.
Comment 4:
The description of modifications to the DeepVariant pipeline is insufficiently detailed. The manuscript states that train.py and keras_modeling.py were modified, but does not specify what exact changes were made. Clearer documentation of the implementation differences is necessary to ensure reproducibility.
Comment 5:
Although 95% confidence intervals are provided, the confidence intervals for the two models overlap, and no statistical significance testing is performed. The observed improvement in SNP F1 score (+0.09%) may fall within the range of random variation. Statistical tests such as the Mann–Whitney U test, paired t-test, or bootstrap-based significance assessment should be considered to validate whether the performance differences are meaningful.
Author Response
Dear Reviewer,
We thank you for your careful evaluation of our manuscript, your detailed and thoughtful review, and your insightful comments. Below we respond to each comment in detail and describe the corresponding revisions implemented in the updated manuscript.
Comment 1:
Regarding reproducibility, the original DeepVariant training pipeline uses seven samples from different individuals and incorporates multiple chromosomes to avoid overfitting. In contrast, the present study reduces the dataset to only two samples. The manuscript does not sufficiently justify why this downscaling was necessary, why these specific two samples were selected, and whether such selection introduces potential bias. It is also unclear whether this reduced dataset is representative of large-scale scenarios and whether the conclusions generalize across diverse populations and sequencing conditions.
Response 1: We appreciate the reviewer's concern regarding dataset scale compared to production grade DeepVariant pipeline. Our primary aim was to evaluate whether replacing the Inception-V3 backbone improves DeepVariant’s accuracy. Training full production-scale models for every architectural candidate is computationally prohibitive; therefore, we adopted a reduced GIAB configuration consistent with DeepVariant prototyping guideline. Sample selection prioritized high-confidence, well-characterized data representative of standard sequencing conditions (NovaSeq 30× WGS). Particular chromosomes were selected because their size and variant density provide a realistic representation of genome-wide variant-calling challenges. Their use allows meaningful architectural comparisons while maintaining computational feasibility. Importantly, we performed full-genome testing on two additional samples—HG003 and HG005—representing distinct ancestries (Ashkenazi Jewish and East Asian, respectively), covering all chromosomes except those used for training/validation. This ensured complete separation of training and testing regions and increased diversity in the evaluation cohort.
We added the following information into Table 5 and 6 “4.1 Datasets” section:
In our study, we followed the same methodological principles but in a reduced configuration. Training is the most resource-demanding stage, and training full production-scale models for each architectural candidate is computationally prohibitive; therefore, we adopted a simplified GIAB setup consistent with the DeepVariant advanced training tutorial (https://github.com/google/deepvariant/blob/r1.6.1/docs/deepvariant-training-case-study.md) to rapidly prototype and evaluate proposed models. In designing the dataset, we aimed to ensure diverse samples, complete separation between training, validation, and testing regions. We used BAM files from the original article’s supplementary materials, kindly provided by the DeepVariant authors [36], corresponding to NovaSeq 30× whole-genome sequencing, a standard and well-characterized coverage for WGS applications (sample details in Table 5). For model development, HG001 chromosome 10 and HG001 chromosome 20 were used for training and validation, respectively. Chromosomes 10 and 20 were selected because they provide representative genome regions of moderate size and variant density, making them suitable for rapid iteration during model development while still reflecting the characteristics of whole-genome variant calling tasks. For the final evaluation, we used the whole HG003 and HG005 samples, excluding the aforementioned regions to prevent data leakage.
Comment 2:
The Introduction lacks adequate comparison with recent advances in neural network architectures. For instance, modern CNNs such as ConvNeXt and other contemporary vision backbones are not discussed. A more thorough review of related work is needed to contextualize the contribution.
Response 2: We have addressed this comment by expanding the Introduction to include a detailed comparison with recent advances in neural network architectures. Specifically, we have incorporated discussions of modern convolutional neural networks such as ConvNeXt and other state-of-the-art vision backbones.
We added the following information into “1. Introduction” section:
Beyond these advancements, computer vision models have increasingly embraced hybrid and fully transformer-based designs. Integration of architectural insights from transformers exemplified by modernized CNNs like ConvNeXt [25, 26]. Vision Transformers (ViT) demonstrated that, with sufficient data, self-attention can rival or surpass convolutional architectures for image classification [27]. Subsequent refinements - such as DeiT [28], which introduced more data-efficient training strategies, and Swin Transformer, which added hierarchical feature maps and shifted windows - facilitated the transition of transformers into domain requiring fine-grained, spatially aware predictions [29].
Comment 3:
The manuscript only compares ‘Inception V3’ with ‘EfficientNet-B3’, making it difficult to support claims of generalizability. It would be helpful to include a comparison table summarizing model size, number of parameters, training time, and performance across different architectures to provide a more comprehensive evaluation.
Response 3: We have updated the manuscript to address this concern by including an additional Table 4 that summarizes model size, number of parameters and training time. Performance metrics across multiple neural network architectures are reflected in Table 1-3.
Comment 4:
The description of modifications to the DeepVariant pipeline is insufficiently detailed. The manuscript states that train.py and keras_modeling.py were modified, but does not specify what exact changes were made. Clearer documentation of the implementation differences is necessary to ensure reproducibility.
Response 4: We have revised the manuscript to provide a clearer and more detailed description of the modifications made to the DeepVariant pipeline.
We added the following information into “4.3.1. Training pipeline adaptation” section:
Key modifications include the introduction of a configurable _BACKBONES dictionary within the model codebase, enabling selection among multiple neural network architectures - ‘Inception V3’ and ‘EfficientNet B0-B7’ - accessed via the tf.keras.applications module, with the potential for future extension to other architectures. The get_model() function was adapted to instantiate the chosen backbone architecture, defaulting to ‘Inception V3’, and configured to accept input tensors representing pileup image data. Additionally, logging was updated throughout to improve training reproducibility.
Comment 5:
Although 95% confidence intervals are provided, the confidence intervals for the two models overlap, and no statistical significance testing is performed. The observed improvement in SNP F1 score (+0.09%) may fall within the range of random variation. Statistical tests such as the Mann–Whitney U test, paired t-test, or bootstrap-based significance assessment should be considered to validate whether the performance differences are meaningful.
Response 5: We have addressed this important comment by performing statistical significance testing using paired t-tests and additionally calculating Cohen's d effect sizes between the compared models (at 2.1.2, 2.2 Table 1-3 ).
We added the following information into Tables 1-3 and “2.1.2 Performance evaluation” section:
The ‘EfficientNet-B3’ model achieves a higher overall F1 score of 95.31%, representing a 0.51 percentage-point improvement over the baseline 94.80% with a significant improvement (paired t-test p=0.0067) and a large effect size (Cohen’s d=1.71). The alternative model achieves a precision of 95.78%, surpassing the strong baseline of 95.63% by 0.15 percentage points; however, this difference is not statistically significant (p = 0.243, Cohen's d = 0.40). Most notably, recall improves to 94.78%, marking a substantial 1.02 percentage-point gain over the baseline 93.76% with a notably large effect size (p = 0.0003, Cohen's d = 1.79).
.. and “2.2. Testing” section:
For SNP both models demonstrate very high performance with F1 scores exceeding 99% (Table 2 and 3). On the never-seen samples - belonging to a distinct ancestry with a divergent genotype class distribution - performance improvement of alternative model persists. Across both HG003 and HG005, the DeepVariant pipeline incorporating ‘EfficientNet-B3’ consistently improves performance. For HG003, the model yields a 0.10-percentage-point increase in F1 (p = 0.0107, Cohen’s d = 1.68), accompanied by a 0.08-point gain in precision (p = 0.0471, Cohen’s d = 0.94) and a 0.11-point increase in recall (p = 0.1708, Cohen’s d = 0.69). Similarly, for HG005, F1 improves by 0.07 percentage points (p = 0.0075, Cohen’s d = 1.75), precision by 0.08 points (p = 0.0372, Cohen’s d = 1.00), and recall by 0.07 points (p = 0.2423, Cohen’s d = 0.57). Together, these results demonstrate consistent gains across metrics, with statistically significant improvements in F1 and precision for both samples and modest non-significant enhancements in recall.
For indel classification - which has traditionally posed a more challenging scenario - both models demonstrate slightly modest performance, with F1 scores approaching 95%. For HG003, ‘EfficientNet-B3’ yields a modest 0.23-percentage-point improvement in F1 (p = 0.6301, Cohen’s d = 0.27). Precision shows a clearer benefit, increasing by 0.18 points (p = 0.0042, Cohen’s d = 1.61), while recall improves by 0.29 points, though without statistical significance (p = 0.7455, Cohen’s d = 0.18). A similar pattern is observed for HG005. F1 improves by 0.19 percentage points (p = 0.3877, Cohen’s d = 0.46), precision increases by 0.23 points with strong statistical support (p = 0.0022, Cohen’s d = 1.86), and recall shows a small, non-significant gain of 0.15 points (p = 0.7070, Cohen’s d = 0.20).
In the context of indel detection, which is considerably less represented in the training data, the proposed model shows clear improvements in precision; however, gains in recall (and consequently sensitivity) remain challenging. In HG003, for example, recall is nearly 10 percentage points lower than precision, underscoring the difficulty the model faces in recovering true indel variants. Although mean recall and F1 values trend upward across independent runs, the high variability observed for indels prevents drawing confident conclusions about consistent improvements in recall and downstream increases in F1 overall.
Reviewer 2 Report
Comments and Suggestions for Authors
The work by Gurianova et al., provides a redesign of the DeepVariant architecture that offers an improved performance in variant calling for this widely used tool. Overall, the work is interesting and the paper well-written with detailed methods. Nevertheless, the authors need to further test and compare the performance of the old and the new model in a larger number of datasets, perhaps in the context of cancer too (PMID: 40670386).
Further suggestions
The information provided in the github repository could be improved, so that interested users can follow the new model step-by-step.
The link in reference 13 does not seem to work. Please check other links too, and better provide DOIs if available.
Author Response
Dear Reviewer,
We thank you for your careful evaluation of our manuscript, your detailed and thoughtful review, and your insightful comments. Below we respond to each comment in detail and describe the corresponding revisions implemented in the updated manuscript.
Comment 1: The work by Gurianova et al., provides a redesign of the DeepVariant architecture that offers an improved performance in variant calling for this widely used tool. Overall, the work is interesting and the paper well-written with detailed methods. Nevertheless, the authors need to further test and compare the performance of the old and the new model in a larger number of datasets, perhaps in the context of cancer too (PMID: 40670386).
Response 1: We appreciate the reviewer’s feedback and valuable suggestions. In response, we have included an additional sample, HG005 (Table 5, 6), to provide a more comprehensive assessment of model performance and expanding testing across whole samples (only excluding chromosomes used in training to prevent leakage). We thank the reviewer for the suggestion regarding testing in the context of cancer and will consider this valuable direction for future research extensions. Changes marked at “2.2. Testing”, “4.1 Datasets” sections.
Further suggestions
Comment 2: The information provided in the github repository could be improved, so that interested users can follow the new model step-by-step.
Response 2: We have improved the GitHub repository to provide clearer instructions for users to replicate the new model training and evaluation process. This includes detailed setup guides, launch examples, and a comprehensive README with prerequisites and usage commands, enhancing accessibility and reproducibility for interested users.
Comment 3: The link in reference 13 does not seem to work. Please check other links too, and better provide DOIs if available.
Response 3: We have corrected the broken link in reference 13.
Reviewer 3 Report
Comments and Suggestions for Authors
This manuscript presents a redesigned DeepVariant pipeline that replaces the long-standing Inception V3 CNN architecture with EfficientNet-B3, demonstrating consistent improvements in variant calling accuracy, training efficiency, and parameter reduction. The authors provide comprehensive training, validation, and testing using GIAB datasets and show improvements in F1 score for both SNP and indel calling.
This work has significant potential, but important revisions are needed to strengthen scientific rigor and clarity
Major comments:
1: While replacing Inception V3 with EfficientNet-B3 is interesting, the manuscript should more clearly explain: Why this specific architecture is expected to be better suited for pileup-like data (not natural images). How this work compares to others who attempted model replacements (e.g., DeepVariant forks, community models).
2:Authors drastically reduce the dataset to:1 chromosome (10) for training,1 chromosome (20) for validation,1 chromosome (21) for testing, Only two GIAB samples (HG001, HG003).
3:The authors report mean and 95% CI across 10 runs, which is good, but missing:Statistical significance testing (e.g., paired t-test or Wilcoxon).Effect size reporting (e.g., Cohen’s d) Clear justification for number of independent runs (why 10?).
Minor comments:
1: Figure 3 (EfficientNet vs. Inception comparison) is clear but labels are too small. Figures 4–6 should include clearer legends. Please annotate whether curves represent training or validation metrics in titles.
2: Recommend citing recent attempts at modernizing DeepVariant or similar pipelines (2021–2024), including research on alternative architectures for genomic image classification.
Author Response
Dear Reviewer,
We thank you for your careful evaluation of our manuscript, your detailed and thoughtful review, and your insightful comments. Below we respond to each comment in detail and describe the corresponding revisions implemented in the updated manuscript.
Major comments:
Comment 1: While replacing Inception V3 with EfficientNet-B3 is interesting, the manuscript should more clearly explain: Why this specific architecture is expected to be better suited for pileup-like data (not natural images). How this work compares to others who attempted model replacements (e.g., DeepVariant forks, community models).
Response 1: In the revised manuscript we clarify why the EfficientNet family is a suitable replacement for Inception-based architectures in the context of pileup-like representations, why we selected the B3 variant in particular and how this work compares to others who attempted model replacements (e.g., DeepVariant forks, community models).
We added the following information into “4.2 Selection of Alternative Model for Demonstration” section:
Among the wide range of alternative architectures - including already mentioned popular choices - we focused on ‘EfficientNet’ family. ‘EfficientNet’ models employ a compound-scaling strategy that jointly increases network depth, width, and input resolution in a balanced and principled manner. This design achieves substantially better accuracy-parameter trade-offs than earlier architectures, including ‘Inception V3’. For pileup tensors - which are smaller and structurally more constrained than natural images - such efficiency is particularly advantageous, enabling the model to capture relevant variation patterns without incurring unnecessary depth or computational cost. Additionally, variant-calling signals frequently appear as subtle local features (base-quality gradients for example), and ‘EfficientNet’s’ MBConv blocks with squeeze-and-excitation mechanisms enhance the network’s sensitivity to these fine-grained cues, even when the available global context is limited.
Within the ‘EfficientNet’ family, we selected the mid-sized ‘EfficientNet-B3’ model (Figure 5). Model capacity and computational cost grow rapidly across the ‘EfficientNet’ scale indices (B0–B7) due to the compound-scaling strategy, and preliminary experiments on public benchmark ImageNet dataset indicated that larger variants (B4–B7) provide only modest accuracy improvements while incurring disproportionately higher memory usage and training time. In contrast, ‘EfficientNet-B3’ offered a favorable balance: it delivered substantial accuracy gains over ‘Inception V3’, while maintaining a model size and inference cost that are compatible with practical DeepVariant training pipeline. This combination of improved accuracy and moderate resource demands made ‘EfficientNet-B3’ a compelling and practically viable candidate for integration into DeepVariant.
.. and “1. Introduction” section:
Several community efforts have experimented with alternative architectures, but have typically focused on domain-specific modifications and yet none provide a direct, head-to-head comparison against DeepVariant’s original ‘Inception V3’ model within the full training pipeline [30,31].
Comment 2:Authors drastically reduce the dataset to:1 chromosome (10) for training,1 chromosome (20) for validation,1 chromosome (21) for testing, Only two GIAB samples (HG001, HG003).
Response 2: We thank the reviewer for highlighting this point. We have expanded the manuscript to clarify that our initial proof-of-concept used a minimal dataset due to computational constraints.
We added the following information into Table 5 and 6 “4.1 Datasets” section:
“In our study, we followed the same methodological principles but in a reduced configuration. Training is the most resource-demanding stage, and training full production-scale models for each architectural candidate is computationally prohibitive; therefore, we adopted a simplified GIAB setup consistent with the DeepVariant advanced training tutorial (https://github.com/google/deepvariant/blob/r1.6.1/docs/deepvariant-training-case-study.md) to rapidly prototype and evaluate proposed models. In designing the dataset, we aimed to ensure diverse samples, complete separation between training, validation, and testing regions. We used BAM files from the original article’s supplementary materials, kindly provided by the DeepVariant authors [36], corresponding to NovaSeq 30× whole-genome sequencing, a standard and well-characterized coverage for WGS applications (sample details in Table 5). For model development, HG001 chromosome 10 and HG001 chromosome 20 were used for training and validation, respectively. Chromosomes 10 and 20 were selected because they provide representative genome regions of moderate size and variant density, making them suitable for rapid iteration during model development while still reflecting the characteristics of whole-genome variant calling tasks. For the final evaluation, we used the whole HG003 and HG005 samples, excluding the aforementioned regions to prevent data leakage.”
.. also we have conducted comprehensive testing across all chromosomes (excluding those used in training) and added a third sample, HG005, covering diverse ancestries.
We added the following information into “2.2. Testing” section:
For SNP both models demonstrate very high performance with F1 scores exceeding 99% (Table 2 and 3). On the never-seen samples - belonging to a distinct ancestry with a divergent genotype class distribution - performance improvement of alternative model persists. Across both HG003 and HG005, the DeepVariant pipeline incorporating ‘EfficientNet-B3’ consistently improves performance. For HG003, the model yields a 0.10-percentage-point increase in F1 (p = 0.0107, Cohen’s d = 1.68), accompanied by a 0.08-point gain in precision (p = 0.0471, Cohen’s d = 0.94) and a 0.11-point increase in recall (p = 0.1708, Cohen’s d = 0.69). Similarly, for HG005, F1 improves by 0.07 percentage points (p = 0.0075, Cohen’s d = 1.75), precision by 0.08 points (p = 0.0372, Cohen’s d = 1.00), and recall by 0.07 points (p = 0.2423, Cohen’s d = 0.57). Together, these results demonstrate consistent gains across metrics, with statistically significant improvements in F1 and precision for both samples and modest non-significant enhancements in recall.
For indel classification - which has traditionally posed a more challenging scenario - both models demonstrate slightly modest performance, with F1 scores approaching 95%. For HG003, ‘EfficientNet-B3’ yields a modest 0.23-percentage-point improvement in F1 (p = 0.6301, Cohen’s d = 0.27). Precision shows a clearer benefit, increasing by 0.18 points (p = 0.0042, Cohen’s d = 1.61), while recall improves by 0.29 points, though without statistical significance (p = 0.7455, Cohen’s d = 0.18). A similar pattern is observed for HG005. F1 improves by 0.19 percentage points (p = 0.3877, Cohen’s d = 0.46), precision increases by 0.23 points with strong statistical support (p = 0.0022, Cohen’s d = 1.86), and recall shows a small, non-significant gain of 0.15 points (p = 0.7070, Cohen’s d = 0.20).
In the context of indel detection, which is considerably less represented in the training data, the proposed model shows clear improvements in precision; however, gains in recall (and consequently sensitivity) remain challenging. In HG003, for example, recall is nearly 10 percentage points lower than precision, underscoring the difficulty the model faces in recovering true indel variants. Although mean recall and F1 values trend upward across independent runs, the high variability observed for indels prevents drawing confident conclusions about consistent improvements in recall and downstream increases in F1 overall.
Comment 3:The authors report mean and 95% CI across 10 runs, which is good, but missing:Statistical significance testing (e.g., paired t-test or Wilcoxon).Effect size reporting (e.g., Cohen’s d) Clear justification for number of independent runs (why 10?).
Response 3: The choice of 10 independent runs balances computational feasibility with statistical power for detecting meaningful differences in variant calling metrics, following common practices. Additionally we have now added paired t-test results and Cohen's d effect sizes to validate statistical significance of performance differences between models, reported alongside means and 95% CIs across 10 CV runs (added into Tables 1-3 and “2.1.2 Performance evaluation” and “2.2. Testing” sections).
Minor comments:
Comment 4: Figure 3 (EfficientNet vs. Inception comparison) is clear but labels are too small. Figures 4–6 should include clearer legends. Please annotate whether curves represent training or validation metrics in titles.
Response 4: We have improved the figures as per the reviewer’s suggestions. In Figure 3, the labels have been enlarged for better readability. For Figures 4–6, we added super titles to distinguish whether curves represent training or validation metrics.
Comment 5: Recommend citing recent attempts at modernizing DeepVariant or similar pipelines (2021–2024), including research on alternative architectures for genomic image classification.
Response 5: We thank the reviewer for this recommendation. We have updated the manuscript to cite recent DeepVariant modernization efforts from 2021–2024. These references provide context for our contributions in genomic image classification.
We added the following information into “1. Introduction” section:
This makes DeepVariant one of the widely used applications employing image-based data representations together with deep learning models for genome analysis, alongside approaches such as Chaos Game Representation (CGR) image-encoded genome analysis and integrative spatial gene-expression imaging models [14].
Several community efforts have experimented with alternative architectures, but have typically focused on domain-specific modifications and yet none provide a direct, head-to-head comparison against DeepVariant’s original ‘Inception V3’ model within the full training pipeline [30,31].
Round 2
Reviewer 2 Report
Comments and Suggestions for Authors
The authors have addressed my comments and the quality of the manuscript has been improved.
Reviewer 3 Report
Comments and Suggestions for Authors
I have no more questions